

# Barley yield and malt quality affected by fall and spring planting under rainfed conditions

Ibrahim Saygili

Field Crops Department, Tokat Gaziosmanpasa University, Tokat, Turkey

## ABSTRACT

**Background**. As a result of the changing climate characteristics, it is necessary to reevaluate the planting time for crop plants. The aim of the present study was to determine the quality characteristics of malting barley cultivars in fall and spring plantings.

**Methods**. Sixteen malting barley cultivars were used. Two fall-planted and two spring-planted trials were conducted in two consecutive years. The field trials were carried out in a randomized complete block design with four replications in Tokat province of Turkey under rainfed conditions.

**Results**. Grain yields varied between 4.38 and 5.71 t/ha in fall-planted trials and between 3.12 and 4.89 t/ha in spring-planted trials. Malt extracts were between 77.0% and 78.0% kg in fall-planted trials and between 73.9% and 76.9% in spring-planted trials. Alpha amylase activities ranged from 77.9 to 81.4 Ceralpha unit (CU)/g in fall-planted trials and from 80.8 to 100.9 CU/g in spring-planted trials. Diastatic power ranged from 194.5 to 331.1 Windisch-Kolbach unit (°WK) in fall-planted trials and from 129.0 to 259.8 °WK in spring-planted trials. GGE biplot analysis indicated that winter barley cultivar Durusu and facultative barley cultivar Ince-04 were the best with consistent grain yields while Ince-04 was the best with stable malt extract across the trials. In scatter plot graphics, winter barley cultivars Durusu, Aydanhanim, Yildiz and facultative Ince-04 had superior performance in fall-plantings for grain yield and malt extract. In spring planting, facultative Ince-04 had higher performance than those of other cultivars. In spring plantings, facultative or winter barley cultivars that do not have strong vernalization requirement had better yield and malt quality. Appropriate planting time and cultivars can allow a better use of available water in malt barley production under rainfed conditions. Lastly, instead of evaluating the grain yield or malt quality alone, it would be best to evaluate the target product (malt extract percentage) obtained from a particular region, process, or production methodology.

## INTRODUCTION

Barley grain is the most preferred source of malt due to its hulls covering the grain, high starch content and good levels of starch-digesting enzymes. Economic return of malting barley is higher than feed barley (*Windes et al., 2019*). Malting barley should be produced in

Corresponding author
Ibrahim Saygili,
ibrahimsaygili50@gmail.com

appropriate ecologies using superior cultivars. Identifying higher performance for special ecologies and appropriate practices would contribute to the production of good quality malting barley.

Barley can be planted in late fall (October-November) or in late winter-early spring (February-March) (hereafter fall and spring plantings, respectively). The winter cultivars that can be planted in late fall should have vernalization requirement along with short-day (<12 h) photoperiod response and winter hardiness (*Locatelli et al., 2022*). The facultative cultivars that can be planted both before and after the winter have no need for vernalization requirement, but should have short-day photoperiod sensitivity and winter hardiness (*Munoz-Amatriain et al., 2020*). Fall plantings generally have higher yield potential than spring plantings (*Windes et al., 2019*) because of longer growing periods and more water availability under rainfed conditions. Although yield potential of fall plantings is high, spring planting could be necessary when the planting could not be realized in fall and in regions where winter are too harsh for barley or in regions which do not receive enough rain for soil preparation in fall. Therefore, performance of winter and facultative malt barley cultivars should be evaluated in spring and fall plantings for malting barley production.

Malt is a product obtained by breaking down the starch by enzymes in the germinated and roasted grain. The most commonly used grain for malt production is barley. The primary indicator of malt quality in barley is the malt extract (*Hoyle et al., 2020*). Another indicator of malt quality is diastatic power. Diastatic power refers to the total activity of enzymes (alpha and beta amylase and limit dextrinase) that convert starch into simple sugars (*Charmier, McLoughlin & McCleary, 2021*). Test weight and thousand-seed weight are also important characters for malt quality. They give indirect information about starch and protein contents of grain (*Kumar et al., 2022*). Although higher test weight and thousand-seed weight are preferred in malting barley, malt extract is the major determinant of malting quality. There is no information about important malt quality traits (malt extract, diastatic power and alpha amylase) of winter and facultative cultivars in fall and spring plantings.

The quantity and quality of the malt obtained from the barley grain is determined by the cultivars used. They are also affected by the growing conditions. Therefore, the quality of the malt is shaped by the interaction of genotype and environment. Various methods are used to determine suitable genotypes and environments. Additive main effects and multiplicative interaction (AMMI) and genotype plus genotype environment interaction (GGE) biplot analysis are widely preferred (*Bilate Daemo et al., 2023*). Using the GGE biplot analysis, *Ghazvini et al. (2022)* examined 20 barley genotypes and identified superior genotypes and their stability. *Yan et al. (2007)* stated that genotype and genotype environment interaction should be used together for the evaluation of a genotype in terms of a trait, and reported that GGE biplot analysis is more advantageous than AMMI for both genotype and genotype environment interaction. Therefore, the use of GGE biplot analysis for the evaluation of genotype and genotype by environment interaction is more useful for better presentation and evaluation of data.

In semi-arid regions, planting time is highly dependent on rains in growing periods. Fall planting would allow better grain fill resulting in plumper grain with lower protein, which

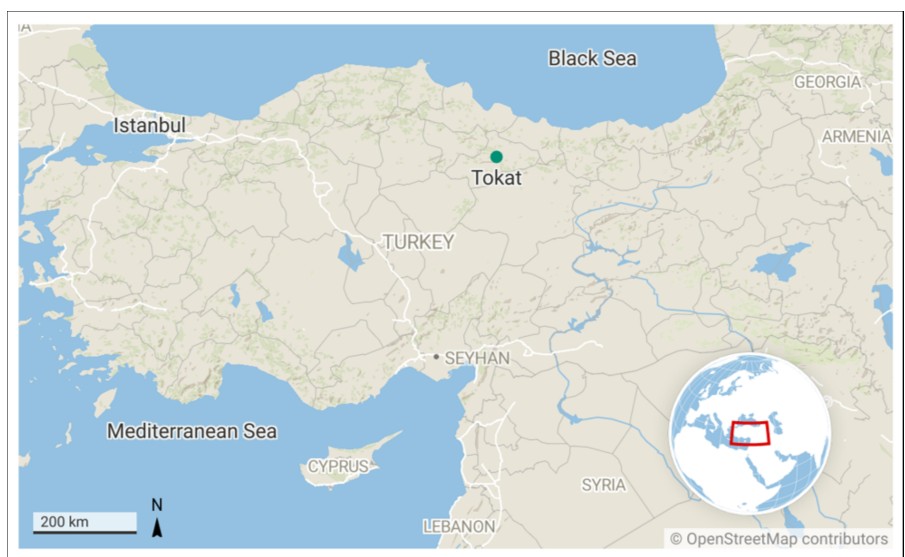

**Figure 1** **Map of the experimental sites.** The map was created at https://app.datawrapper.de/map/ygawh/basemap.

is favorable for malt barley. But planting after winter could also be necessary. The aim of the present study was to determine the quality characteristics of malting barley cultivars in fall and spring plantings.

## MATERIAL AND METHODS

### Plant materials
Field trials were carried out in the experimental fields of Tokat Agricultural Application and Research Center of Tokat Gaziosmanpasa University (40°33′N, 36°47′ E, 539 m a.s.l) in 2015/16 and 2016/17 in Tokat province of Turkey (Fig. 1) under rainfed conditions. Seven winter, eight facultative and a spring malting barley cultivar Harrington were examined (Table 1).

### Field trials
Experimental soils had clayed loam texture, slight levels of salt (0.041 and 0.042%), mild alkaline reaction (pH: 7.77 and 7.80), moderate amounts of lime (10.2 and 7.7%), high and low amounts of $P_2O_5$ (111 and 58 kg/ha), high levels of $K_2O$ (1108 and 1015 kg/ha) and low levels of organic matter (1.45 and 1.22%) in 2015/16 and 2016/17 years, respectively (Table 2) (Soil analyses were carried out by soil laboratory of the Middle Black Sea Transitional Zone Agricultural Research Institute, Tokat-Turkey). The average temperature and monthly total precipitation of November-June period for long term (48 years) and experimental years are given in Fig. 2. The average long-term precipitation was 359 mm, and the precipitation in 2016 (315 mm) was slightly less than the long term while the precipitation in 2017 (229 mm) was considerably less than the long-term average. The average precipitation of the spring-planting trials was 195 mm in long term (March–June).

**Table 1** Origin, pedigree, and growth habits of the barley cultivars used.

| Cultivar | Origin | Institute | Pedigree | Growth Habit |
|---|---|---|---|---|
| Aydanhanim | Turkey | FCCRI | GK Omega/Tarm 92 | Winter |
| Basgul | Turkey | AEBM | Severa/Tokak//Ad. Gerste/Clipper | Facultative |
| Bolayir | Turkey | TARI | Osk 4.197/12-84//HB854/Astrix/3/Alpha/Durra | Winter |
| Catalhoyuk | Turkey | AEBM | S 8602/Kaya | Winter |
| Cumra-2001 | Turkey | AEBM | Tokak selection/Beka | Winter |
| Durusu | Turkey | AEBM | W9013/Kaya//Severa | Winter |
| Efes-98 | Turkey | AEBM | Tercan selection/Tipper | Facultative |
| Erciyes | Turkey | AEBM | Severa/Tokak//Ad. Gerste/Clipper | Facultative |
| Harrington | Canada | MSUS | Klages/3/Gazelle/Betzes//Centenial | Spring |
| Ince-04 | Turkey | AARI | 4671/Tokak//4648/p12-119/3/WBCB-4 | Facultative |
| Kalayci-97 | Turkey | AARI | Erginel 90//364 TH/Tokak | Facultative |
| Ozdemir-05 | Turkey | AARI | CUM/4060//P12-62/P169-2 | Facultative |
| Sladoran | Croatia | TARI | Introduction from Croatia (Alpha/Mursa) | Winter |
| Tokak 157/37 | Turkey | FCCRI | Selection from Turkish landraces | Facultative |
| Yildiz | Turkey | AEBM | Angore//S8602/Clarine | Winter |
| Zeynelaga | Turkey | FCCRI | Anteres/KY63-1249//Lignee131 | Facultative |

Notes.
FCCRI, Field Crops Central Research Institute; TARI, Thrace Agricultural Research Institute; AEBM, Anadolu Efes Bira and Malt Co.; AARI, Anatolian Agricultural Research Institute; MSU, Montana State University, University of Saskatchewan.

**Table 2** Soil characteristics and planting dates of the field trials.

| Year | Soil texture | Total salt % | pH | CaCO$_3$ % | P$_2$O$_5$ kg/ha | K$_2$O kg/ha | Organic matter % | Planting dates in fall | Planting dates in spring |
|---|---|---|---|---|---|---|---|---|---|
| 2015/16 | Clayed-loam | 0.041 | 7.77 | 10.2 | 111 | 1108 | 1.45 | 10 Nov 2015 | 14 Mar 2016 |
| 2016/17 | Clayed-loam | 0.042 | 7.80 | 7.7 | 58 | 1015 | 1.22 | 16 Nov 2016 | 27 Feb 2017 |

In 2016, the precipitation (211 mm) was similar to that in long term while the precipitation in 2017 (139 mm) was considerably less. According to these data, it would not be wrong to consider 2016 as a high rainfall environment and 2017 as a low rainfall environment. The trials were carried out in randomized complete block design with four replications. Each plot consisted of five rows of 4 m long. Row spacing was 20 cm. The seeding rate was 500 plants m$^2$. Fertilizers were applied to plots as 80 kg/ha P$_2$O$_5$ (triple super phosphate) and 80 kg/ha N (Ammonium nitrate).

## Measured traits

Time to heading was the period from sowing to first awn occurrence in 50% of the plants in the plot (*Saygili & Kandemir, 2021*). Plant height was the distance between the ground and the top spikelet in spike except for awn in 15 random plants, and lodging is a visual estimation of the plants lodged in the plot (*Kandemir et al., 2000*). Maturity was the period from sowing to the time when all leaves turned to yellow (*Kandemir et al., 2022*). Number of spikes per square meter was calculated by dividing the grain yield (g/m$^2$) by number of grains per spikes and thousand-seed weight (*Saygili & Kandemir, 2021*). Number of grains

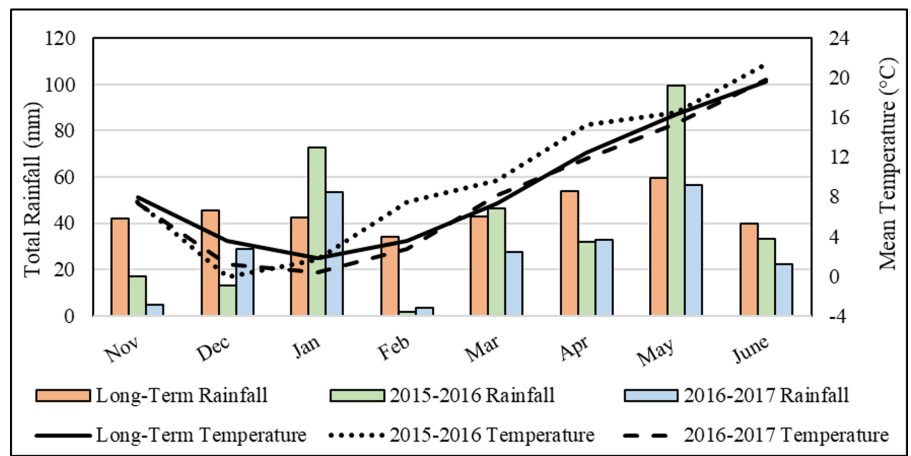

**Figure 2** Temperature and precipitation values in the region for long term and experimental years.

per spikes, grain yield, test weight and thousand-seed weight were determined according to *Aisawi et al. (2015)*. Spike length and number of grains per spikes were determined in 30 randomly selected spikes from the plots. Grain yield was determined by converting the grain product obtained from 4 m² plots to t/ha. Thousand-seed weight was determined by counting and weighing 400 random grains. Test weight was determined by weighing the free-falling grains into an exact volume of 250 ml and converting them to hectoliters. Moisture content of grain was determined by drying for 48 h at 75 °C. Grain yield, test weight and thousand-seed weight were calculated on 12% moisture content basis (*Kandemir et al., 2022*).

Malt production was performed as described by *Saygili et al. (2021)*. Grain was immersed in two cycles soaking, nine hours each, in water and 16 h of draining. Grains were germinated at 14 °C for 110 h. Moisture level was maintained at 45% of the starting grain weight. When the plumula grew to 75% of the grain length, germination was ended. Kilning was performed in consecutive steps of eight hours at 60 °C, six hours at 70 °C and five hours at 80 °C. The rootlets were manually removed. Malt extract was determined using the method described by *Fox & Henry (1993)* with some modifications. A total of 12 ml (approach by weighing) of distilled water at 65 °C was added to 3 g of ground malt passed through a 0.5 mm sieve. The mixture was incubated in a 65 °C water bath for 60 min and was centrifuged for 5 min at 3000 g. The supernatant was measured with a refractometer. Malt extract was determined according to the formula:

$$\text{Malt extract}(\%) = \frac{([\text{sample}(3g) + \text{amount of water}] * \text{refractometer measurement} * 100)}{\% \text{dry matter}}.$$

Diastatic power was measured using the method described by *Fox et al. (1999)*. A total of 10 ml of extraction solution (0.5% NaCl) was added to 1 g of malt and vortexed. The mixture was incubated in a water bath at 25 °C for 30 min. It was centrifuged for 5 min at 2,000 g. A total of 5 ml of buffered starch solution (2% starch, 2 mM glacial acetic acid and 0.05 M sodium acetate, pH: 4.6) was pre-incubated at 25 °C for 5 min. The enzyme

extraction supernatant (100 μl) was added to the buffered starch solution and incubated exactly for 10 min at 25 °C and then 1 ml of 0.5 M sodium hydroxide was added to stop the reaction. Five ml of PAHBAH (p-hydroxy benzoic acid hydrazide (5 g/l) dissolved in alkaline diluent, 0.05 M trisodium citrate, 0.01 M calcium chloride, 0.5 M sodium hydroxide) was pre-incubated in a boiling water bath for at least five min. One hundred μl of hydrolysed sample was added to incubated PAHBAH solution and was kept in boiling water for exactly four min. Content is rapidly cooled to room temperature in slurry ice. It was diluted 10-fold and measured at absorbance at 415 nm. Maltose equivalents are determined according to maltose standards curve absorbances (0, 1, 2, 3, and 4 mg/l) at 415 nm. Diastatic power was determined as Windisch-Kolbach unit (°WK) with the following formula. $°WK = (87.5 \times \text{maltose equivalent}) - 16$.

Alpha amylase activity was analyzed using a commercial kit (Megazyme International Ireland Limited, Product code: K-CERA, Wicklow, Ireland) based on manufacturer's instructions. Alpha amylase activity was expressed as Ceralpha units (CU). CU could be converted to dextrinizing unit ASBC method (DU) and AACC method (SKB) with the following formulas: $DU = 0.23 \times \text{Ceralpha units} + 0.61$ and $SKB \text{ units} = 0.42 \times \text{Ceralpha units} - 0.34$, respectively.

### Statistical analyses

Since variances of years and planting date were not homogeneous based on Barlett's homogeneity test ($p > 0.05$), variance analyses of years and planting date were performed separately (*Saygili et al., 2021*; *Bilate Daemo et al., 2023*) using JMP Pro 14 software (SAS Institute Inc., Cary, NC, USA). Differences between means were grouped by Tukey multiple comparison test ($p < 0.05$). GGE biplot and stability analysis were conducted using the GEA-R software (*Pacheco et al., 2015*) according to *Kandemir et al. (2022)* and *Kandemir (2022)* respectively. To compare cultivars for two traits, a scatter plot graphic was drawn using Minitab (ver.17). PCA-Biplot was used for trait-based scaling of cultivars separately for fall and spring trials.

## RESULTS

### Time to heading

A total of four trials, two fall-planted and two spring-planted, were conducted to determine the performance of the malting barley cultivars. Since cultivar Cumra-2001 produced very few spikes in spring-planted trials, its data in the spring-planted trials were not included in the analyses. The heading time of the cultivars ranged from 163.5 to 178.3 days and from 69.9 to 88.8 days in fall- and spring-planted trials, respectively (Table 3). In fall-planted trials, Cumra-2001 (184.7 and 166.0 days in 2016 and 2017, respectively) and Aydanhanim (187.3 and 171.0 days in 2016 and 2017, respectively) were the latest in heading, while Zeynelaga (173.7 and 155.5 in 2016 and 2017, respectively) and Bolayir (174.3 and 157.5 days in 2016 and 2017, respectively) were the earliest ($p < 0.05$). In spring-planted trials, Aydanhanim (82.7 and 104.8 days in 2016 and 2017, respectively) reached to heading in significantly longer periods than other cultivars. Unlike the fall-planted trials, Sladoran had late heading in spring-planted trials (78.6 and 106.5 days in 2016 and 2017, respectively).

Although Durusu, Yildiz, Bolayir and Sladoran reached to heading early in fall-planted trials, their headings were late in spring-planted trials.

## Time to maturity

Time to maturity varied between 199.7 and 227.4 days in fall-planted trials and between 111.9 and 113.1 days in spring-planted ones in 2016 and 2017, respectively (Table 3). In both fall-planted trials, Aydanhanim (210.8 and 231.7 days), Cumra-2001 (211.5 and 232.3 days) and Durusu (210.5 and 230.0 days) matured latest, while Catalhoyuk (192.5 and 223.0 days), Efes-98 (191.5 and 223.0 days), Erciyes (192.5 and 225.0 days) and Ozdemir-05 (193.3 and 225.0 days) matured earliest ($p < 0.05$ days). In spring-planted trials, Aydanhanim matured late in both years. While Durusu, Harrington, Sladoran and Yildiz were among the cultivars that matured late in the 2016 trial, in 2017 they matured somewhat early. In spring-planted trials, Catalhoyuk was earliest in 2016S (105.7 days), while in 2017S Basgul (103.5 days), Efes-98 (105.5 days) and Zeynelaga (103.5 days) matured relatively earlier than other cultivars. Sladoran, Yildiz and Zeynelaga matured consistently late in 2016S trial and early in 2017S trial.

## Plant height

In fall-planted trials, there was a difference of 24 cm in average plant height between 2016 and 2017 trials (84.9 and 109.2 cm, respectively) while in spring-planted trials the difference was only 4.2 cm. In fall-planted trials, Aydanhanim, Basgul, Cumra-2001 and Efes-98 had consistently taller plants (Table 3, $p < 0.05$). Catalhoyuk, Erciyes, Ince-04 and Tokak 157/37 had taller plants in 2016F trial but relatively moderate plant heights in 2017F trial. Sladoran had consistently low plant heights (68.0 and 88.1 cm) in the two fall-planted trials. In 2016S, Aydanhanim had the tallest plants (98.8 cm), while in 2017S, Basgul, Catalhoyuk, Efes-98, Ince-04 and Tokak 157/37 had taller plants (88.3–95.6 cm). Sladoran and Zeynelaga had shorter plants in the two spring-planted trials.

## Lodging

In fall-planted trials, lodging was 46.5 and 67.7% in 2016 and 2017, while in spring-planted it was 47.2 and 8.5% in 2016 and 2017, respectively (Table 4). In fall- and spring-planted trials, Basgul, Catalhoyuk, Efes-98, Erciyes, Kalayci-97, Ozdemir-05 and Tokak 157/37 had consistently higher lodging values ($p < 0.05$). On the other hand, Bolayir, Cumra-2001, Durusu, Sladoran, Yildiz and Zeynelaga had the lowest lodging values.

## Number of grains per spike

Number of grains per spike was similar in all trials except for 2017F. The highest numbers of grains in fall-planted trials were obtained from Aydanhanim (30.3 and 30.5) and Cumra-2001 (29.5 and 30.8) (Table 4, $p < 0.05$). All cultivars other than Basgul, Bolayir, Durusu, Harrington, Sladoran and Yildiz had the lowest number of grains per spike in fall-planted trials. In spring-planted trials, the highest number of grains per spike was obtained from Aydanhanim (29.5 and 28.9) and Harrington (27.4 and 28.0) in the two years. All cultivars except for Yildiz (27.3 and 24.7) had less seeds per spike (20.5–25.4) in spring-planted trials.

**Table 3 Effect of cultivar and planting date on heading time, maturity and plant height.**

| Cultivars | Heading time (day) | | | | Maturity (day) | | | | Plant height (cm) | | | |
|---|---|---|---|---|---|---|---|---|---|---|---|---|
| | Fall | | Spring | | Fall | | Spring | | Fall | | Spring | |
| | 2016 | 2017 | 2016 | 2017 | 2016 | 2017 | 2016 | 2017 | 2016 | 2017 | 2016 | 2017 |
| Aydanhanim | 187.3a | 171.0a | 82.7a | 104.8ab | 231.7a | 210.8a | 116.0a | 129.3a | 94.3a | 121.8a | 98.8a | 81.9def |
| Basgul | 176.7def | 164.8bcd | 65.0ef | 77.5h | 227.7b-e | 189.5e | 112.7b | 103.5g | 93.4a | 114.8abc | 94.4b | 93.9ab |
| Bolayir | 174.3fg | 157.5fg | 78.3b | 101.0c | 224.3efg | 203.0bcd | 111.7b | 119.3c | 67.4f | 112.1b-e | 79.6f | 71.0hi |
| Catalhoyuk | 179.0bcd | 165.8bc | 66.3c-f | 79.0h | 223.0g | 192.5e | 105.7f | 106.5f | 93.2a | 113.3bcd | 94.1b | 91.2abc |
| Cumra-2001 | 184.7a | 166.0b | – | – | 232.3a | 211.5a | – | – | 94.5a | 114.9abc | – | – |
| Durusu | 179.7bc | 164.5bcd | 78.0b | 103.8b | 230.0abc | 210.5a | 115.0a | 123.5b | 82.2c | 104.0ef | 90.6bcd | 72.7ghi |
| Efes-98 | 177.7b-e | 165.8bc | 66.0def | 77.5h | 223.0g | 191.5e | 108.0de | 105.5fg | 92.7a | 116.1ab | 93.8b | 90.3a-d |
| Erciyes | 177.3cde | 165.3bc | 65.7ef | 79.0h | 225.0d-g | 192.5e | 108.0de | 106.5f | 90.0ab | 109.3b-e | 88.3cde | 84.1c-f |
| Harrington | 180.3b | 164.3bcd | 68.0cd | 87.8d | 227.3c-f | 199.8d | 114.7a | 113.5d | 75.5de | 97.3f | 93.0b | 80.7efg |
| Ince-04 | 177.3cde | 165.8bc | 67.0cde | 85.8e | 228.0bcd | 200.5cd | 113.0b | 113.5d | 89.8ab | 110.4b-e | 90.6bcd | 88.3a-e |
| Kalayci-97 | 179.0bcd | 163.8cd | 68.0cd | 82.9f | 224.0fg | 199.8d | 107.7e | 112.8d | 84.2c | 109.2b-e | 91.2bc | 86.3b-f |
| Ozdemir-05 | 176.3efg | 164.5bcd | 64.7f | 81.0g | 225.0d-g | 193.3e | 109.3cd | 107.3f | 85.9bc | 105.8de | 84.4e | 82.2def |
| Sladoran | 176.0efg | 158.0f | 78.7b | 106.5a | 230.8abc | 205.5b | 115.7a | 121.3c | 68.0f | 88.1g | 76.9fg | 65.2i |
| Tokak 157/37 | 179.3bcd | 162.8de | 66.0def | 84.3ef | 225.0d-g | 201.3bcd | 109.7c | 110.3e | 93.8a | 113.2bcd | 91.2bc | 95.6a |
| Yildiz | 174.7fg | 161.3e | 68.3c | 101.5c | 231.0ab | 204.6bc | 115.3a | 119.6c | 80.6cd | 110.3b-e | 86.8de | 73.3ghi |
| Zeynelaga | 173.7g | 155.5g | 65.0ef | 79.3gh | 230.3abc | 189.5e | 115.3a | 103.5g | 71.9ef | 106.7cde | 74.0g | 79.3fgh |
| Mean | 178.3 | 163.5 | 69.9 | 88.8 | 227.4 | 199.7 | 111.9 | 113.1 | 84.9 | 109.2 | 88.6 | 84.2 |

**Notes.**

–, data could not be obtained. Means followed by a common letter are not significantly different by the Tukey test at the 5% level of significance..

Saygili (2023), *PeerJ*, DOI 10.7717/peerj.15802

**Table 4  Effect of cultivar and planting on lodging, number of grains per spike and spikes per square meter.**

| Cultivars | Lodging (%) | | | | Number of grains per spike | | | | Number of spikes per square meter | | | |
|---|---|---|---|---|---|---|---|---|---|---|---|---|
| | Fall | | Spring | | Fall | | Spring | | Fall | | Spring | |
| | 2016 | 2017 | 2016 | 2017 | 2016 | 2017 | 2016 | 2017 | 2016 | 2017 | 2016 | 2017 |
| Aydanhanim | 16.7c | 71.3abc | 6.3d | 0.0b | 30.3a | 30.5a | 29.5a | 28.9a | 423.4fgh | 381.7fi | 277.3g | 303.0bcd |
| Basgul | 93.3a | 97.5a | 100.0a | 20.0ab | 21.3f | 27.0bc | 22.1de | 21.7de | 507.7bcd | 344.7hi | 440.6cde | 287.0cd |
| Bolayir | 0.0d | 42.5cd | 0.0d | 0.0b | 21.0f | 26.6bc | 25.4bc | 24.9b | 567.5ab | 390.8e-i | 494.6bc | 311.4bcd |
| Catalhoyuk | 100.0a | 100.0a | 100.0a | 35.0a | 22.7def | 25.7b-e | 21.6de | 22.0de | 482.2c-f | 359.7ghi | 383.2ef | 304.7bcd |
| Cumra-2001 | 2.0d | 43.8cd | – | – | 29.5a | 30.8a | – | – | 373.3gh | 373.5f-i | – | – |
| Durusu | 0.0d | 42.5cd | 0.0d | 0.0b | 25.3bc | 25.9b-e | 25.3bc | 22.5cd | 435.2efg | 497.3abc | 461.0bcd | 296.0cd |
| Efes-98 | 100.0a | 100.0a | 100.0a | 12.5ab | 22.7def | 24.6cde | 21.9de | 23.2bcd | 494.4cde | 327.8i | 288.1g | 276.3d |
| Erciyes | 100.0a | 100.0a | 100.0a | 15.0ab | 22.9def | 26.0b-e | 21.3de | 20.1e | 459.7def | 389.3e-i | 518.5b | 350.1ab |
| Harrington | 10.0c | 47.5cd | 0.0d | 0.0b | 26.4b | 26.5bcd | 27.4ab | 28.0a | 537.7abc | 475.9abc | 498.2bc | 272.4d |
| Ince-04 | 26.7b | 46.3cd | 13.8c | 0.0b | 22.8def | 25.3b-e | 22.5de | 22.1de | 532.6abc | 499.9ab | 531.5b | 305.6bcd |
| Kalayci-97 | 96.7a | 77.5ab | 100.0a | 10.0ab | 22.2ef | 25.0c-e | 21.6de | 21.2de | 489.4cde | 509.8a | 409.5de | 304.2bcd |
| Ozdemir-05 | 100.0a | 97.5a | 100.0a | 0.0b | 21.0f | 23.9e | 21.0de | 22.6cd | 593.3a | 429.2c-g | 477.5bcd | 363.7a |
| Sladoran | 0.0d | 48.8bcd | 1.3d | 0.0b | 23.7cde | 25.3b-e | 23.4cd | 23.4bcd | 493.8cde | 454.9a-e | 407.9de | 329.7abc |
| Tokak 157/37 | 98.7a | 100.0a | 90.0b | 35.0a | 21.8ef | 23.7e | 21.1de | 22.4d | 366.7h | 402.0d-h | 381.0ef | 291.3cd |
| Yildiz | 0.0d | 25.0d | 0.0d | 0.0b | 24.5bcd | 27.6b | 27.3ab | 24.7bc | 429.1e-h | 434.6b-f | 327.1fg | 207.8e |
| Zeynelaga | 0.0d | 25.5cd | 0.0d | 0.0b | 21.8ef | 24.1de | 20.5e | 21.9de | 547.5abc | 470.5a-d | 610.9a | 331.5abc |
| Mean | 46.5 | 67.7 | 47.4 | 8.5 | 23.7 | 26.2 | 23.5 | 23.3 | 483.3 | 421.4 | 433.8 | 302.3 |

**Notes.**

–, data could not be obtained. Means followed by a common letter are not significantly different by the Tukey test at the 5% level of significance..

### Number of spikes per square meter

The number of spikes per square meter ranged from 421.4 to 483.3 in fall-planted trials and from 302.3 to 433.8 in spring-planted trials (Table 4). In fall-planted trials, the highest number of spikes per square meter was obtained from Harrington as 537.7 and 475.9, from Ince-04 as 532.6 and 499.9 and from Zeynelaga as 547.5 and 470.5 in 2016 and 2017, respectively ($p < 0.05$). In addition, Bolayir and Ozdemir-05 produced more spikes per square meter in 2016F trial and Durusu, Kalayci-97 and Sladoran produced more spikes in 2017F trial. Cumra-2001 had the lowest number of spikes per square meter in both years. Zeynelaga produced more spikes per area in the two spring-planted trials, while Erciyes, Ozdemir-05 and Sladoran produced more spikes than other cultivars in 2017S trial only. Yildiz had the lowest number of spikes per square meter in the two spring-planted trials, while Aydanhanim and Efes-98 in the 2016S trial only.

### Thousand-seed weight

The thousand-seed weight of cultivars varied between 39.7 and 50.6 g in fall-planted trials and between 44.7 and 49.0 in spring-planted trials (Table 5). Durusu had higher thousand-seed weights in the two fall-planted trials (57.7 and 45.1 g in 2016 and 2017, respectively), while Tokak 157/37 had higher values in 2016F trial (59.3 g), and Zeynelaga (46.1 g) and Yildiz (44.6 g) had higher values in 2017F trial ($p < 0.05$). In the two spring-planted trials, Durusu (52.0 and 48.5 g), Ince-04 (52.0 and 50.7 g) and Tokak 157/3 (53.6 and 50.4 g) had larger seeds while Basgul (50.4 g) and Zeynelaga (49.4 g) produced higher thousand-seed weights in the 2017S trial.

### Test weights

In fall-planted trials, Aydanhanim (66.7 kg), Cumra-2001 (67.5 kg), Ince-04 (66.7 kg), Yildiz (66.6 kg) and Zeynelaga (67.3 kg) had higher test weights (Table 5, $p < 0.05$). Ince-04 had higher test weights in the two spring-planted trials (66.1 and 65.3 kg in 2016 and 2017, respectively). The highest test weight was obtained from Zeynelaga(66.9 kg) in the 2017S trial.

### Grain yields

Grain yields varied between 4.38 and 5.71 t/ha in fall-planted trials and between 3.12 and 4.89 t/ha in spring-planted trials (Table 5). Aydanhanim (6.88 t/ha) and Ince-04 (6.42 t/ha) had higher grain yields in 2016F trial while Durusu (5.79 t/ha) and Yildiz (5.33 t/ha) in 2017F trial ($p < 0.05$). Durusu (6.07−3.23 t/ha), Ince-04(6.19−3.42 t/ha) and Zeynelaga (5.75−3.58 t/ha) had higher grain yields in the two spring-planted trials. In the GGE biplot of grain yield, where PC1 was 67.05 and PC2 was 22.8, Durusu, Ince-04, Erciyes, Tokak 157/37, Efes-98 and Aydanhanim were genotypes located at the extremes (Fig. 3A). Durusu and Ince-04 were the best cultivars in fall-planted trials. There was no environment where other cultivars were best in terms of genotype and genotype x environment interaction. In Fig. 3B, where stability and average yields were compared, the line shown by a circled arrow is average-environment coordination. The arrow indicates the direction in which the means were highest (*Frutos, Galindo & Leiva, 2014*). The cultivars with the highest yield averages were Durusu, Ince-04, Zeynelaga, Harrington, Aydanhanim, Bolayir and Yildiz.

Saygili (2023), *PeerJ*, DOI 10.7717/peerj.15802

**Table 5  Effect of cultivar and planting on thousand seed weight, test weight and grain yield.**

| Cultivars | Thousand seed weight (g) | | | | Test weight (kg) | | | | Grain yield (t/ha) | | | |
|---|---|---|---|---|---|---|---|---|---|---|---|---|
| | Fall | | Spring | | Fall | | Spring | | Fall | | Spring | |
| | 2016 | 2017 | 2016 | 2017 | 2016 | 2017 | 2016 | 2017 | 2016 | 2017 | 2016 | 2017 |
| Aydanhanim | 53.8c | 43.1b | 51.3bc | 35.7e | 66.7ab | 63.2cde | 61.9cd | 55.7c | 6.88a | 5.01bcd | 4.19e | 3.13abc |
| Basgul | 49.9de | 40.0c | 51.4bc | 50.4a | 61.9fg | 59.9fgh | 62.7bcd | 64.9ab | 5.40gh | 3.72g-h | 5.00cd | 3.14abc |
| Bolayir | 47.1f | 40.0c | 43.8f | 41.5d | 64.4cd | 64.4bc | 60.4f | 61.3b | 5.60e-h | 4.15fg | 5.50bc | 3.22abc |
| Catalhoyuk | 47.1f | 36.4def | 51.0bc | 45.6b | 60.0hi | 57.7hi | 61.7de | 62.1b | 5.14ghi | 3.37hi | 4.23e | 3.05abc |
| Cumra-2001 | 55.8b | 43.0b | – | – | 67.5a | 61.6def | – | – | 6.14bcd | 4.95b-e | – | – |
| Durusu | 57.7a | 45.1ab | 52.0ab | 48.5a | 64.8c | 63.9cd | 63.0bcd | 63.1ab | 6.36bc | 5.79a | 6.07ab | 3.23abc |
| Efes-98 | 49.8de | 37.6d | 50.8bc | 44.0bc | 61.6fgh | 57.3i | 63.0bcd | 62.3ab | 5.60e-h | 3.03i | 3.21f | 2.82bcd |
| Erciyes | 48.7def | 34.9fg | 49.7c | 44.6bc | 59.6i | 58.9ghi | 62.3bcd | 63.9ab | 5.12hi | 3.52hi | 5.46bc | 3.15abc |
| Harrington | 42.2g | 35.2efg | 40.2g | 42.9cd | 62.8def | 60.7efg | 63.0bcd | 63.3ab | 5.99b-e | 4.43ef | 5.48bc | 3.26abc |
| Ince-04 | 53.1c | 40.5c | 52.0ab | 50.7a | 66.7ab | 63.4cd | 66.1a | 65.3ab | 6.42ab | 5.10bcd | 6.19a | 3.42ab |
| Kalayci-97 | 50.3d | 37.2def | 51.5bc | 44.4bc | 62.7ef | 60.3fgh | 62.6bcd | 64.3ab | 5.47fgh | 4.74cde | 4.56de | 2.86bc |
| Ozdemir-05 | 43.4g | 33.8g | 44.8ef | 42.9cd | 60.8ghi | 59.5f-i | 62.4bcd | 63.5ab | 5.40gh | 3.47hi | 4.50de | 3.54a |
| Sladoran | 48.1ef | 40.2c | 46.8d | 35.9e | 64.3cd | 63.7cd | 63.2bc | 61.8b | 5.62d-g | 4.61def | 4.46de | 2.77cd |
| Tokak 157/37 | 59.3a | 37.3de | 53.6a | 50.4a | 63.8cde | 58.5ghi | 63.0bcd | 62.5ab | 4.73i | 3.56hi | 4.31e | 3.29abc |
| Yildiz | 53.5c | 44.6ab | 49.8c | 44.0bc | 64.5c | 66.6ab | 60.5ef | 60.8b | 5.61d-g | 5.33ab | 4.45de | 2.26d |
| Zeynelaga | 49.5de | 46.1a | 45.9de | 49.4a | 65.3bc | 67.3a | 63.5b | 66.9a | 5.90c-f | 5.23bc | 5.75ab | 3.58a |
| Mean | 50.6 | 39.7 | 49.0 | 44.7 | 63.6 | 61.7 | 62.6 | 62.8 | 5.71 | 4.38 | 4.89 | 3.12 |

**Notes.**

–, data could not be obtained. Means followed by a common letter are not significantly different by the Tukey test at the 5% level of significance..

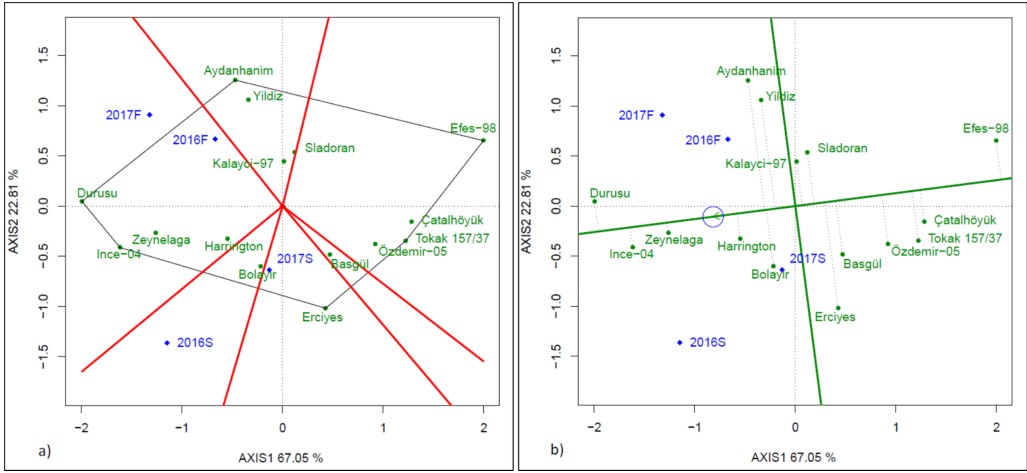

**Figure 3** **GGE biplot (A) and stability *vs* means (B) presenting of grain yield.**

Accordingly, the cultivars closest to the average yield line are the most stable ones (*Frutos, Galindo & Leiva, 2014*). The most stable cultivars with above the average grain yields were the Zeynelaga, Ince-04 and Durusu. Aydanhanim, Yildiz and Bolayir, on the other hand, were unstable.

## Alpha amylase activity

Alpha amylase activities ranged from 77.9 to 81.4 CU/g in fall-planted trials and from 80.8 to 100.9 CU/g in spring-planted trials (Table 6). Yildiz had the highest Alpha amylase activity in the two fall-planted trials (153.8 and 212.2 CU/g in 2016 and 2017, respectively), and Sladoran (133.7 CU/g) had the highest alpha amylase activity in 2017F trial ($p < 0.05$). In the two spring-planted trials, Harrington(190.1–201.1 CU/g) had the highest alpha amylase values.

## Diastatic power

Diastatic power ranged from 194.5 to 331.1° WK in fall-planted trials and from 129.0 to 259.8° WK in spring-planted trials (Table 6). Harrington (611.1° WK) had the highest diastatic power in the 2016F trial, while in the 2017F trial, Bolayir (372.1° WK) and Durusu (320.4° WK) had the highest values ($p < 0.05$). In spring-planted trials, Bolayir (452.3° WK) had the highest diastatic power in the 2016S trial and Zeynelaga (301.3° WK) in 2017S trial.

## Malt extract

In fall-planted trials, average malt extract was 78.0% in 2016 and 77.0% in 2017, which were 76.9 and 73.9 in spring-planted trials in 2016 and 2017, respectively. In fall-planted trials, Durusu, Harrington and Ince-04 had consistently higher malt extract (Table 6, $p < 0.05$). However, Yildiz had higher malt extract percentage in 2016F trial and Aydanhanim and Bolayir had higher malt extract percentages in the 2017F trial. In spring-planted trials, Harrington (80.7 and 81.9%) and Ince-04 (79.1 and 81.7%) had higher malt extract

Saygili (2023), *PeerJ*, DOI 10.7717/peerj.15802

**Table 6  Effect of cultivar and planting on alpha amylase activity, diastatic power and malt extract.**

| Cultivars | Alpha amylase activity (CU/g) | | | | Diastatic power (°WK) | | | | Malt extract (%) | | | |
|---|---|---|---|---|---|---|---|---|---|---|---|---|
| | Fall | | Spring | | Fall | | Spring | | Fall | | Spring | |
| | 2016 | 2017 | 2016 | 2017 | 2016 | 2017 | 2016 | 2017 | 2016 | 2017 | 2016 | 2017 |
| Aydanhanim | 118.6bc | 99.2bc | 67.0de | 82.1d | 436.4c | 348.6ab | 325.3b | 133.6c | 79.8cde | 80.1ab | 75.3ef | 74.7bc |
| Basgul | 39.2h | 54.9f | 42.9f | 80.0de | 143.6gh | 93.4ghi | 127.3d | 52.4d | 77.2gh | 75.5d | 75.4ef | 72.8cd |
| Bolayir | 85.0de | 93.0b-e | 108.4b | 86.3d | 388.1cd | 372.1a | 452.3a | 147.5c | 80.3bcd | 79.8ab | 79.7bc | 74.3bcd |
| Catalhoyuk | 50.6gh | 42.9f | 70.5cde | 69.2de | 156.4fgh | 121.0fgh | 173.6cd | 74.4d | 78.1e-h | 71.7fg | 74.3fg | 72.3de |
| Cumra-2001 | 46.1gh | 60.0def | – | – | 311.4e | 234.1c | – | – | 76.3hi | 78.5bc | – | – |
| Durusu | 95.2cde | 96.0bcd | 91.6bc | 133.4bc | 359.3de | 320.4ab | 354.4b | 124.8c | 80.7a-d | 80.6ab | 78.7cd | 75.2b |
| Efes-98 | 46.6gh | 58.7ef | 47.0ef | 73.5de | 117.2h | 105.8f-i | 185.6cd | 75.6d | 72.2k | 76.5cd | 74.8f | 72.3de |
| Erciyes | 56.2fgh | 32.6f | 62.6ef | 67.0de | 203.6f | 61.9i | 147.5cd | 46.5d | 77.3fgh | 70.4g | 74.3fg | 69.4fg |
| Harrington | 124.0b | 107.5b | 190.9a | 201.1a | 611.1a | 305.9b | 429.1a | 222.7b | 82.2a | 81.8a | 81.9a | 80.7a |
| Ince-04 | 79.3ef | 54.8f | 90.6bcd | 132.3bc | 434.1c | 157.3def | 295.8b | 255.5b | 81.3abc | 80.0ab | 81.7ab | 79.1a |
| Kalayci-97 | 74.5efg | 48.8f | 55.9ef | 99.9cd | 177.3fg | 78.0hi | 161.4cd | 41.8d | 74.4j | 73.2ef | 72.4g | 68.1g |
| Ozdemir-05 | 43.7h | 67.0c-f | 47.5ef | 78.6de | 190.6fg | 122.3fgh | 204.5c | 58.8d | 74.9ij | 71.7fg | 75.1ef | 70.3ef |
| Sladoran | 133.7ab | 116.5b | 99.3b | 90.9d | 373.5d | 242.1c | 307.5b | 129.1c | 78.4efg | 78.4bc | 79.0cd | 73.7bcd |
| Tokak 157/37 | 49.3gh | 46.7f | 70.3cde | 43.3e | 143.5gh | 148.5efg | 133.7cd | 143.1c | 74.6ij | 75.4de | 73.5fg | 74.8bc |
| Yildiz | 153.8a | 212.2a | 100.5b | 130.9bc | 393.6cd | 213.4cd | 317.7b | 127.6c | 81.7ab | 79.6b | 80.7abc | 74.9bc |
| Zeynelaga | 108.5bcd | 55.5f | 67.3de | 145.3b | 538.7b | 186.8cde | 281.8b | 301.3a | 79.0def | 79.2b | 77.0de | 75.4b |
| Mean | 81.4 | 77.9 | 80.8 | 100.9 | 331.1 | 194.5 | 259.8 | 129.0 | 78.0 | 77.0 | 76.9 | 73.9 |

Notes.

–, data could not be obtained. Means followed by a common letter are not significantly different by the Tukey test at the 5% level of significance. CU, Ceralpha Unit. WK, Windisch-Kolbach unit..

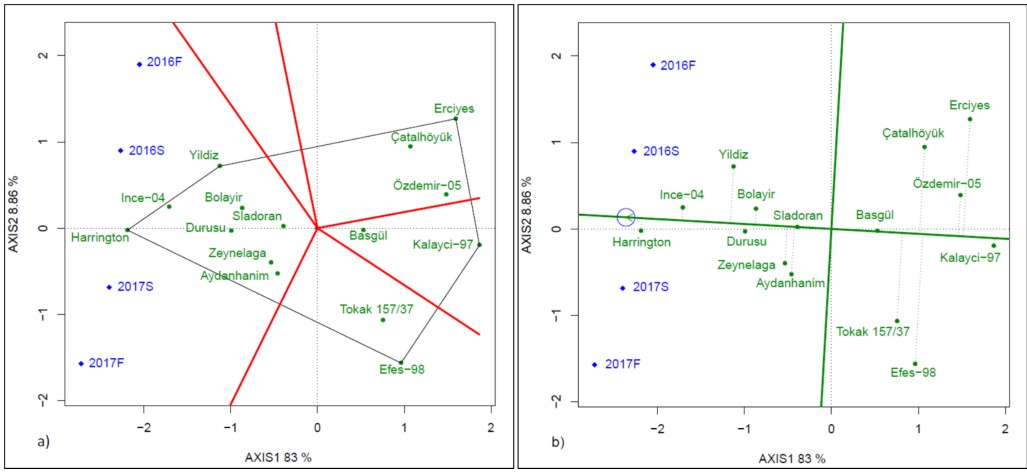

**Figure 4 GGE biplot (A) and stability *vs* means (B) presenting of malt extract percentage.**

percentages in both years, while Yildiz had a higher malt extract percentage only in the 2016S trial. In which-won-where view of the GGE biplot, Harrington, Yildiz, Erciyes, Kalayci-97 and Efes-98 were located at the vertices of the polygon, which indicated that these cultivars had the best or poorest performance in an environment or environments (Fig. 4A). Harrington and Yildiz were the best cultivars in terms of malt extract percentages. Ince-04, Durusu, Bolayir, Sladoran, Zeynelaga and Aydanhanim also performed well in all trials. In Fig. 4B, the cultivars with the highest malt extract percentages in decreasing order were Harrington, Ince-04, Yildiz, Durusu, Bolayir, Zeynelaga, Aydanhanim and Sladoran. Among the cultivars with high malt extract percentages, Harrington, Ince-04 and Durusu were relatively stable ones, while Yildiz and Aydanhanim were unstable ones.

## Evaluation of all traits
A principal component analysis was carried out in order to evaluate all traits. The first two principal components (PC) accounted for 74.23% of the total variation in fall trials and 77.85% in spring trials (Fig. 5). The most predominant characters in fall trials were lodging, grain yield and malt extract on PC1, and lodging, diastatic power and plant height in spring trials. Pattern of lodging in both planting times was associated with less malt quality. Malt extract, alpha amylase, and diastatic power were closely related in both planting time. In the winter trials Yildiz, Ince-04 and Durusu, and in the spring trials Harrington and Durusu had a trend in the same direction with malt quality criteria such as malt extract, diastatic power and alpha amylase activities. Zeynelağa had the same pattern with the number of ears per square meter in both trials. In fall trials, Durusu and Yildiz showed a similar pattern with grain yield, and Ince-04 in spring trials.

## DISCUSSION
The preference of planting time depends on adaptation to changing climates, periods of water availability and winter temperatures. Grain yields of the cultivars in the present study

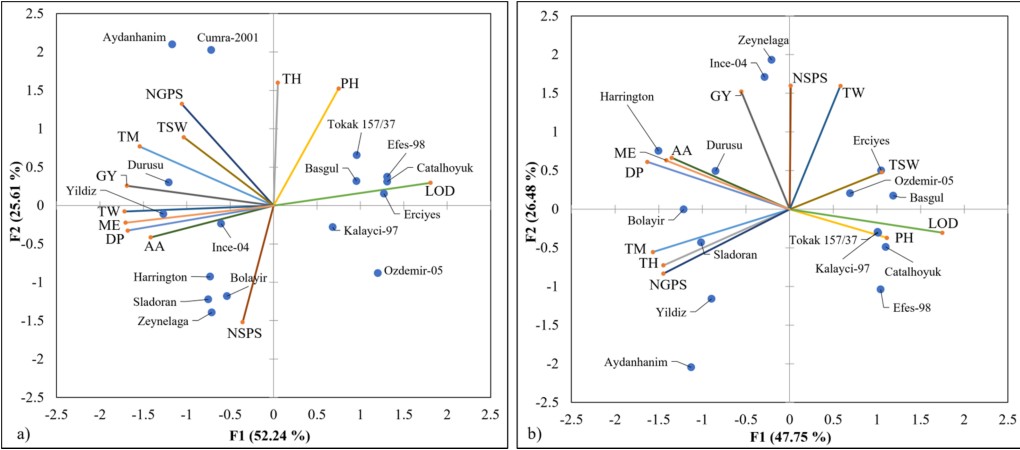

**Figure 5  Principal component analysis in fall (A) and spring plantings (B).** AA, alpha amylase activities; DP, diastatic power; GY, grain yield; LOD, lodging; ME, malt extract; NGPS, number of grains per spikes; NSPS, number of spikes per square meter; PH, plant height; TH, time to heading; TM, time to maturity; TSW, thousand-seed weight; TW, test weight.

were quite high in fall-planted trials. However, the results showed that the spring plantings can also have high yields when the precipitation is sufficient. Nevertheless, fall plantings had advantages such as longer growing periods with higher amount of precipitation, early maturity, more water availability in soil, and more flexibility of planting time than spring plantings. Therefore, barley produced in fall plantings has higher grain yield potential.

The yields of Aydanhanim spring-planted trials were quite low compared to the fall-planted trials. Although late heading and maturity produce higher yields in high rainfall conditions, they can pose a risk in low rainfall or shorter growing conditions such as spring plantings. Cumra-2001, whose grain yields were satisfactory in fall-planted trials, produced very few spikes in spring-planted trials. This showed that cultivar Cumra-2001 has strong vernalization requirement. Similarly, Aydanhanim may have had low yields in spring-plantings due to vernalization requirement. Winter barley cultivars with strong vernalization requirement cannot meet vernalization needs in spring plantings (*Fernández-Calleja, Casas & Igartua, 2021*). Therefore, Cumra-2001 and Aydanhanim should not be preferred in spring plantings. Durusu, Ince-04 and Zeynelaga were notable for higher grain yields under spring plantings. The late heading of Durusu and Yildiz in spring plantings may also be due to the partial vernalization need. Since early heading and maturity are an escape mechanism from drought (*Dorrani-Nejad et al., 2022*; *Kandemir & Saygili, 2023*), consistently early heading cultivars may be preferred in regions with drought risk. The late maturity in 2016S trial compared to 2017S may have been due to higher precipitation in May 2016. Although the maturity was delayed by increased rainfall in vegetation period (*Kassie & Tesfaye, 2019*), cultivars such as Zeynelaga, Yildiz and Ince-04 can adjust their vegetation period in order to better benefit from the rains. Therefore, these cultivars can better adapt to today's changing climatic conditions. With the preference of such cultivars,

high yields can also be obtained in regions where spring-planting is mandatory in semi-arid regions.

Lodging affects yield and quality of the crops negatively. Tolerance to lodging minimizes the development of diseases that adversely affect malting quality and are harmful to humans (*Tidemann et al., 2020*). Short stature is highly correlated with lodging tolerance. Lodging was less in cultivars with shorter plant heights in the present study. However, it was observed that lodging was also low in Aydanhanim and Ince-04 which had relatively taller plants. Apart from plant height, stem elasticity and better root development are also effective in lodging resistance (*Niu et al., 2022*). High precipitation in fall planting, which boosted more grain production, resulted in higher lodging damage than spring planting. However, cultivars with good lodging tolerance also had a high yield potential. Therefore, lodging tolerance further supports grain yields. In fall planting where precipitation is high and lodging is a major problem, short stature, stem elastic and high yielding cultivars should be preferred.

The number of grains per spike, the number of spikes per square meter and thousand-seed weight are the determinants of grain yield, and there is a balance among these characters (*Angassa & Mohammed, 2022*). This balance is very important in malt barley grown in rainfed conditions, because increasing number of spikes per square meter and number of grains per spike may decrease thousand-seed weight and starch content. Therefore, cultivars that can maintain high thousand-seed should be preferred for malt barley production under low precipitation conditions. Zeynelaga produced higher number of spikes per square meter in all trials and had the highest thousand-seed weight in low-rainfall trials. Thousand-seed weight of Durusu was also consistently high in all trials. On the other hand, Basgul, Ince-04, Tokak 157/37 and Zeynelaga had higher thousand-seed weights in 2017S trial, which had the lowest amount of precipitation. High thousand-seed weight under low precipitation conditions is an indicator of drought tolerance (*Kebede, Kang & Bekele, 2019*). Accordingly, drought tolerant barley cultivars can provide sufficient starch for malt under low precipitation conditions.

Malt extract percentage is the best quality criterion of malt barley. In the present study, malt extract was higher in fall-planted trials. *Oziel et al. (1996)* also found that malt extract percentages were higher in fall-planted trials. Malt barley should have 80% or higher malt extract percentage (*Fox et al., 2003*). The highest malt extract was obtained from the Canadian malting barley cultivar Harrington. Aydanhanim, Bolayir, Durusu, Ince-04 and Yildiz also had high malt extract over 80%. Higher malt extract percentage indicates that there are enough starch and enzymes that break down starch. The test weights of these cultivars were high. In fact, test weight was expected to decrease under low precipitation conditions *i.e.,* spring plantings. However, the higher test weights in spring-planted trials may have been due to the obtaining less grains per unit area. Therefore, it can be concluded that in spring-planted trials, enough starch accumulation occurs to achieve high test weights in a small number of grains per unit area. Otherwise, higher values of test weight could not be obtained in spring plantings where the grain-filling period when starch accumulation occurs is always shorter and drier than fall-planted trials. The reason of high malt extract value obtained from Aydanhanim, Bolayir, Durusu and Ince-04 in fall-planted

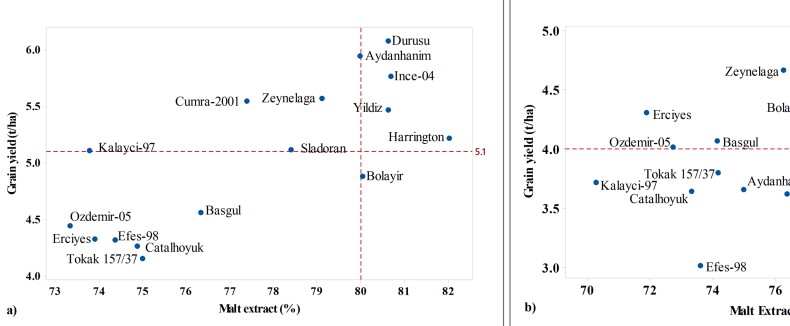

**Figure 6** Scatter plot of grain yields and malt extract of cultivars in fall (A) and spring plantings (B).

trials is that these cultivars may have good activities of malting enzymes other than alpha amylase. The alpha amylase activities of these cultivars except for Yildiz were below the recommended value (150 CU/g) for malting barley (*Fox et al., 2003*). Ince-04 and Yildiz had higher malt extract ratios compared to other cultivars in spring-planted trials. Indeed, their malt extract values in spring-planted trials were as good as those in fall-planted trials. In spring-planted trials, cultivars other than Harrington had lower alpha amylase activities than what was recommended for malt barley. Harrington also had quite high malt extract percentage (*Marquez-Cedillo et al., 2000*), diastatic power (*Igartua et al., 2000*) and beta amylase activity (*Barr et al., 2003*). Therefore, Harrington can be used as donor genotype for the genes to enhance enzyme activities. Lastly, malt barley production should be performed as fall planting with winter or facultative barley cultivars in regions that do not have very hard winters. However, in cases where spring-planting is obligatory, a facultative or winter cultivars with mild vernalization requirement should be preferred.

Stable malting barley cultivars are important for malting industries (malt quality) and producers (grain yield). Therefore, grain yield and malt quality characteristics should be evaluated together in malt barley cultivars. A scatter plot was drawn (Fig. 6) between grain yield and malt extract, the primary indicator of malt quality (*Hoyle et al., 2020*). In Fig. 6, the lower limit of grain yield was the average yield (*Kurt, 2020*) while the lower limit for malt extract was the recommended level of 80% (*Fox et al., 2003*). In the graph, the cultivars above and right of the area determined by the lower limits could be concluded as the cultivars having the best malting performance. Based on the findings, winter Durusu, Aydanhanim, Yildiz and facultative Ince-04 were identified as the best performing cultivars in fall-planted trials (Fig. 6A) and Ince-04 in spring-planted trials (Fig. 6B). Ince-04 stood out in terms of malt quality and grain yield in all environments.

## CONCLUSIONS

Grain yield and malt quality need be evaluated simultaneously for the production of malt barley. Instead of evaluating the grain yield or malt quality alone, it would be best to evaluate the target product (malt extract percentage) obtained from a unit area. Higher grain yield and malt quality were achieved in fall plantings. Winter cultivars Durusu,

Aydanhanim, and Yildiz could be recommended for fall planting. On the other hand, facultative cultivar Ince-04 high performance and good stability, can be recommended also in fall and spring plantings. Therefore, in region where both fall and spring planting could be performed, facultative cultivars, which have no disadvantages in fall and spring planting, would be more appropriate. The present study was conducted in only one location, albeit repeated over the years. For similar ecologies, cultivars and planting time can be preferred according to the results of this research, but in other regions with different precipitation and altitude than those of the present study, further research may be needed. Rather than making the evaluation based on a single trait with GGE biplot, comparative evaluation of the two most important characters (grain yield and malt extract in the present research) makes it easier to present the results. In the light of this information, the GGE biplot, means *vs.* stability, principal coordinate analysis, and the scatter plot graphics provide a more practical presentation of the data obtained in the research.

### Funding
The author received no funding for this work.

### Competing Interests
The author declare that they have no competing interests.

### Author Contributions
- Ibrahim Saygili conceived and designed the experiments, performed the experiments, analyzed the data, prepared figures and/or tables, authored or reviewed drafts of the article, and approved the final draft.

### Data Availability
    The raw data are available in the Supplemental Files.

### Supplemental Information
Supplemental information for this article can be found online at http://dx.doi.org/10.7717/peerj.15802#supplemental-information.

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
