# Peer review of "Barley yield and malt quality affected by fall and spring planting under rainfed conditions"

_PeerJ, doi:10.7717/peerj.15802_

## Round 0.1 · original submission · Major Revisions

Dear Author

The manuscript cannot be accepted for publication in its current form. It needs a major revision to be reconsidered for publication. The author is invited to revise the paper considering all the suggestions made by the reviewers. Please note that requested changes are required for publication.

With Thanks

Reviewer 1 ·

Basic reporting

Literature references, sufficient field background/context provided

Experimental design

Methods should be described with sufficient information to be reproducible by another investigator.

Validity of the findings

nothing

Annotated reviews are not available for download in order to protect the identity of reviewers who chose to remain anonymous.

·

Basic reporting

The study is meeting the scientific standard in terms of technical language, experimental design, and implication of the results. The English language of article seems better, but some typo errors should be checked briefly during revision. I suggest a minor revision for this article and will be happy to review its revised version. The following suggestions should be considered to improve the manuscript quality before its publication in PeerJ.

Experimental design

need more information.

Validity of the findings

The English language of article seems better, but some typo errors should be checked briefly during revision. I suggest a minor revision for this article and will be happy to review its revised version. The following suggestions should be considered to improve the manuscript quality before its publication in PeerJ.

Additional comments

I suggest a minor revision for this article and will be happy to review its revised version. The following suggestions should be considered to improve the manuscript quality before its publication in PeerJ.

·

Basic reporting

The authors used sixteen malting barley cultivars were used. Two fall-planted and two spring-planted trials were conducted in two consecutive years to determine the quality characteristics of malting barley cultivars in fall and spring plantings, which aims to determine the performance of winter and facultative malt barley cultivars based on agronomic and malt quality traits such as ME, diastatic power and alpha amylase activities in fall and spring plantings. He used the Tukey test to determine significant differences between genotypes used and GGE biplot (a) and stability, but this study needs more analyses and improvement to obtain accurate selection criteria such as path analysis, principal component, principal, cluster, additive main effects and multiplicative interaction (AMMI) and broad-sense heritability.
- And also analysis of the interaction between treatments and /or years.
- The Tukey test at the 1% level of significance is the best.
- In the results, separate titles must be placed for each part.

Experimental design

Methods described with sufficient detail & information to replicate and appropriate

Validity of the findings

It was supporting results

---

## Round 0.2 · accepted · Accept

Dear Authors
I am pleased to inform you that after the last round of revision, the manuscript has been improved a lot, and it can be accepted for publication.
Congratulations on the acceptance of your manuscript, and thank you for your interest in submitting your work to Peer j
Best Regards


Reviewer 1 ·

Basic reporting

The manuscript looks good

Experimental design

No comment

Validity of the findings

No comment

·

Basic reporting

accept
all our comments were corrected by authors.

Experimental design

accept
all our comments were corrected by authors.

Validity of the findings

accept
all our comments were corrected by authors.

Additional comments

accept
all our comments were corrected by authors.

·

Basic reporting

All of my concerns have been addressed by the author

Experimental design

All of my concerns have been addressed by the author

Validity of the findings

All of my concerns have been addressed by the author